**Data Availability Statement:** All relevant data are within the manuscript and its Supporting Information files.

**Funding:** This work was supported by The Ministry of Education of the Republic of Korea and the

# Development of an elderly lifestyle profile: A Delphi survey of multidisciplinary health-care experts

Kang-Hyun Park[1,¤a,‡], Ji-Hyuk Park[2,¤b,*]

**1** Department of Occupational Therapy, Sangji University, Wonju, South Korea, **2** Department of Occupational Therapy, Yonsei University, Wonju, South Korea

¤a Current address: Department of Occupational Therapy, Sangji University, Wonju, South Korea
¤b Current address: Department of Occupational Thearpy, Yonsei University, Wonju, South Korea
‡ The author is 1st Author to this work.
* otscientist@yonsei.ac.kr

## Abstract

### Objective

Lifestyle is considered as a key factor that affects one's health and quality of life, and it has become the focus of increasing research interest worldwide. Research has shown that life-style is an important health determinant in preventive health care. However, lifestyle is a multifaceted construct and there is limited evidence regarding lifestyle assessment, which evaluates individuals' multi-domain lifestyle factors. This study developed evaluation items for measuring the multifaceted lifestyle profile of community-dwelling older adult to prevent chronic disease and improve their health and quality of life.

### Methods

Opinions from 21 experts with experience in older adults and lifestyle research were col-lected from December 2019 to January 2020. Three Delphi surveys were carried out, based on previous research. The first survey gathered opinions using a mix of open- and closed-ended questions regarding items of the older adults' multifaceted lifestyle profile. The sec-ond was conducted after adding and modifying several items based on the first Delphi sur-vey. In the third survey, after the results of the second one were presented to the expert panels, final opinions from the experts were converged.

### Results

In total, 59 items were selected as the first Delphi results; 62 items were selected as the sec-ond results after adding and modifying the values below a content validity ratio of .42; and 62 items were selected as the third Delphi results. The average content validity ratio of the final Delphi survey was .92, the stability was .18, and the consensus was .80, which were all high.

National Research Foundation of Korea NRF-2018S1A3A2074904 to J-HP.

**Competing interests:** The authors have declared that no competing interests exist.

## Conclusions

This study verified the content validity of the evaluation items for community-dwelling older adults' multifaceted lifestyle profile. In the future, it is expected that after verifying the validity and reliability, this will be used as a standardized assessment tool in clinical environments.

## Introduction

According to World Population Prospects 2019 [1], there were 703 million people aged 65 years or over in the world in 2019. The number of older adults is projected to double to 1.5 billion by 2050. Population aging is a human success story; however, the increasing longevity tends to increase the financial burden on each country [1]. As populations age, it is essential to ensure continued and equitable access to disease prevention, treatment, and rehabilitation during all stages of life. Healthy aging is more than just the absence of disease; it also entails the maintenance of functional ability throughout the lifespan [2]. Therefore, health and long-term care systems need to be aligned to meet the needs of aging populations by providing age-appropriate lifestyle interventions based on accurate lifestyle profiling.

Occupational therapy is focused on a preventive approach that considers individuals' different lifestyle factors and daily routines that are related to health and quality of life [3]. Previous research demonstrated that lifestyle redesign programs and the modification of lifestyle factors have been successful in promoting functional capacities and well-being among community-dwelling older adults [4] [5] [6]. Yet, the challenge of multifaceted lifestyle profiling of the older adults remains [7] [8]. Only a few measurements evaluate one's health-related lifestyle; these tend to only include dietary factors, physical exercise, smoking, and/or drinking habits [9]. While lifestyles are important to improving the health and well-being of older adults, there has been little specific measurement of individuals 'multifaceted lifestyle.

Lifestyle is considered a key factor that affects one's health and quality of life, and it has become the focus of increasing research interest worldwide [10]. According to the World Health Organization, approximately 60% of an individual's health-related quality of life depends on their lifestyle [2]. Moreover, recent evidence from lifestyle research on older adults demonstrated that a healthy lifestyle reduces the incidence of major chronic diseases such as diabetes and heart disease, as well as disability and mortality rates [11] [12] [13] [14]. Research also shows that lifestyle is an important health determinant in preventive health care. Although there is copious evidence on lifestyle interventions, the existing lifestyle research focuses on diet, physical activity, and unhealthy habits such as drinking and smoking as components and indicators of lifestyle [7]. However, lifestyle is a multifaceted construct. According to a review of the literature [4], a comprehensive definition of lifestyle for older adults mentions how a person spends their time and money, which represents an older adult's values and personality. A review of the literature further identified a need to address this issue, as there are few specific measurements to assess a older adult's lifestyle.

The goal of the present research was to develop a multifaceted lifestyle profile for older adults, agreed upon by multidisciplinary health-care experts including occupational therapists, physiotherapists, nutritionists, doctors, nurses, and social workers, in order to improve the validity of the profile. The intended use of this profile is as a screening tool to allow the quick identification of older adults who have unhealthy lifestyles and who need a more in-depth assessment to determine their fitness to live in the community independently.

## Methods

### Study design

We used a modified Delphi survey technique, a methodology commonly used to obtain a group consensus among "experts" through a series of structured open and closed questions [15] [16]. Ethics approval was granted by Yonsei University (YUWIRB-1041849-201901-BM-012-01).

### Participants

We aimed to recruit health professionals deemed experts in the lifestyles and health of the older adults. For this, the researchers invited health professionals who had completed a gradu-ate master's degree or who had experience in relevant areas for over five years.

To enhance the diversity of the panelists' expertise, we also targeted professionals from South Korea with recognized expertise in the fields of older adults' lifestyles and/or health-care management for older adults within the scope of occupational therapy, physiotherapy, nutri-tion, social welfare, and preventive medicine. The eligibility criteria were to (a) be able to com-plete three rounds of the Delphi survey within a two-month timeframe, (b) be competent in using a computer, and (c) have access to the Internet and email.

Informed consent was obtained for the Delphi survey. Potential participants were sent an invitation email containing a letter introducing the survey and written consent form regarding the research. The participants checked a consent tick box on the first page of the survey. Partic-ipants submitted the informed written consent through email. Participants had the option to not complete the survey during any point in the process. To compensate them for their time, all participants received a $50 gift card after completing all rounds of the survey.

### Delphi method procedures

After obtaining their consent, sociodemographic information was collected from the respon-dents. Participants were asked to read about the concept of multifaceted lifestyle components that were developed for creating a lifestyle profile of older adults in the first page of the survey questionnaire attached to an email (S1 Appendix in S1 Data). In the first page, physical activ-ity, nutrition and activity participation are described as key domains of lifestyle based on the literature review and previous study [9]. Participants were then shown the next page which was consisted of 62 questions: 19 questions in the "physical activity" domain, 21 in the "activity participation" domain, and 22 in the "nutrition" domain. These items were developed by the researchers in accordance with our previous studies [4] [9] [17].

The Delphi survey consisted of three rounds (Fig 1). After completing the initial step, the panel participants were e-mailed an invitation to activate the round 1 questionnaire. The research team provided information regarding the study and received informed consent from all participants on the expert panel before starting the study. The participants were allowed to start the round 1, 2, and 3 questionnaire, which was delivered via email. The surveys for each round were available for two weeks, and a reminder was sent one day after the due date to par-ticipants who had not responded in order to maximize the response rate.

### Round 1

The round 1 survey consisted of 59 closed-ended questions and 3 open-ended questions grouped into 3 themes. The survey was developed with open and closed questions to accom-modate expert opinions. Participants were encouraged to add any recommendations or

**Development of the Questionnaire**

**Stage 1**

- Literature review
- Item extraction from previous studies
- Item categorization and restructuring

?

**Expert Delphi Survey**

**Round 1 Delphi survey (n=21)**

- 21 expert panel selection
- 62 open- and closed-ended questions grouped into three themes

**Round 2 Delphi survey (n=21)**

- Survey on the relevance of each item that was extracted from the first Delphi survey
- Several items were added and modified

**Stage 2**

**Round 3 Delphi survey (n=21)**

- Several items were modified
- Analysis of the survey items to determine the content validity rate, agreement, stability, and convergence

?

**Development of multifaceted lifestyle profile for the elderly**

**Stage 3**

Final 62 items were derived

**Fig 1. Research process.**

opinions regarding the questionnaire. The round 1of the survey required about 30 to 40 minutes to complete.

## Round 2

The round 2 survey was developed based on the participants' responses in round 1. The round 2 survey consisted of 62 closed-ended questions grouped into 3 categories (physical activities,

**Table 1. Example question asked to assess the relevance of each element in a particular category.**

| Category 1: Physical activity in your lifestyle | |
| --- | --- |
| **Items** | **Relevance** |
| Participation in aerobic exercise | Strongly irrelevant 1 2 3 4 Strongly relevant |
| Participation in anaerobic exercise | Strongly irrelevant 1 2 3 4 Strongly relevant |
| Participation in low-intensity exercise | Strongly irrelevant 1 2 3 4 Strongly relevant |
| Participation in moderate-intensity exercise | Strongly irrelevant 1 2 3 4 Strongly relevant |
| Participation in high-intensity exercise | Strongly irrelevant 1 2 3 4 Strongly relevant |
| Participation in walking exercise | Strongly irrelevant 1 2 3 4 Strongly relevant |

activity participation, and nutrition). The participants received the survey via e-mail and were required to score the relevance of each proposed element using a four-point Likert-type scale (strongly relevant, relevant, not relevant, or strongly irrelevant). A neutral middle point was not included to force participants to provide their obvious opinions (Table 1). In round 2, three items were added from the round 1 items and these items were composed and grouped intro 3 categories.

## Round 3

Round 3 modified four questions and no questions from the second round were excluded. For 62 items, 80% agreement was reached. In the third round, we required the participants to rate the relevance of each item using a four-point Likert-type scale ranging from 1 (strongly irrelevant) to 4 (strongly relevant). The level of consensus was set to 80% of respondents indicating agreement [18] [20].

## Data analysis

The round 1 result was subjected to a content analysis to identify and classify common elements in the overall opinion of the expert panel. We performed categorization by adding sub-items and modifying unclear expressions. In order to analyze the round 2 and 3 results, we calculated content validity ratios (CVRs). The minimum CVR was determined by the number of experts participating in each round [19]. According to the criteria, the CVR values of all items were set to 0.42 for 21 panel experts in rounds 2 and 3. The stability, which means the panel's agreement on each item, was analyzed by the coefficient of variation divided by the arithmetic mean of the standard deviation of each item. If the coefficient of variation is less than 0.5, no further Delphi investigation is required, and if it is 0.5–0.8, it is relatively stable [20]. Regarding consensus, a higher consensus score demonstrated a higher level of agreement between participants [20].

## Results

### Demographics of the panel experts

We sent out a total of 30 invitations to potential participants. Of these, 21 participants gave their written consent to participate. All of 21 experts who participate in this study completed the round 1, 2, and 3 survey. There was no one who drop out the survey in the three steps. The characteristics of the final sample (n = 21) are presented in Table 2. Thirteen participants (62%) were female. Eighteen participants (86%) had over eight years' experience in their professional area. The panel consisted of experts from various health-related professions, such as

**Table 2. Demographic characteristics of the respondents (*n* = 21).**

| | | Round 1 N (%) | Round 2 N (%) | Round 3 N (%) |
|---|---|---|---|---|
| Sample size | | 21 | 21 | 21 |
| Response rate | | 100% | 100% | 100% |
| Gender | Male | 8 (38%) | 8 (38%) | 8 (38%) |
| | Female | 13 (62%) | 13 (62%) | 13 (62%) |
| Work experience | 5–7 years | 3 (14%) | 3 (14%) | 3 (14%) |
| | 8–10 years | 9 (43%) | 9 (43%) | 9 (43%) |
| | 11 ≤ years | 9 (43%) | 9 (43%) | 9 (43%) |
| Occupation (clinical/research/both) | Occupational therapist | 4 (19%) | 4 (19%) | 4 (19%) |
| | Nutritionist | 4 (19%) | 4 (19%) | 4 (19%) |
| | Social worker | 3 (14%) | 3 (14%) | 3 (14%) |
| | Physiotherapist | 4 (19%) | 4 (19%) | 4 (19%) |
| | General practitioner | 1 (5%) | 1 (5%) | 1 (5%) |
| | Nurse | 4 (19%) | 4 (19%) | 4 (19%) |
| | Researcher | 1 (5%) | 1 (5%) | 1 (5%) |

occupational therapists, nutritionists, social workers, physiotherapists, general practitioners, nurses, and researchers.

## Results of round 1

The round 1 results are described in detail in Table 3. Twenty-one participants accessed the round 1survey and answered all questions. Round 1 was composed of 62 questions including 59 closed-ended questions and 3 open-ended questions in 3 groups. For the physical activity and activity participation categories, consensus was reached on all items. However, in terms of nutrition category, in order to measure the specific type of lifestyle of the older adults, the expert panel suggested three new items (nut intake, fruit intake, and dairy product intake). Thus, after completing the round 1, three new items were proposed.

## Results of round 2

The categories and items on the Delphi panel survey are listed in Table 4. In the physical activity, activity participation, and nutrition domains of the multifaceted older adults' lifestyle, the CVR was 0.42 or higher and the content validity was verified for all items. The average value and CVR were the highest in "aerobic physical exercise," "anoxic physical exercise," "walking," "leisure activities," "social activities," "education," and "rest and sleep." Contrariwise, the CVR for "low-intensity physical activity" was the lowest. However, the CVR scores were 0.61, 0.52, and 0.52 for the sub-themes of low-intensity activity; thus, these sub-items were not excluded. Thus, following the second round, all items and sub-items reached consensus. It is relevant to highlight that none of the initially proposed items were rejected during round 2.

## Results of round 3

None of the 62 items analyzed had a minimum CVR of 0.42 or less (Table 5). As a result of round 3, compared with the second round, the relevance of the items was relatively high, and the panel's responses resulted in relatively high convergence and agreement (Table 6).

**Table 3. Categories and items of first Delphi results.**

| Category | Items |
|---|---|
| Physical activity | Aerobic physical exercise |
| | Anoxic physical exercise |
| | High-intensity physical exercise |
| | Moderate-intensity physical exercise |
| | Low-intensity physical activity |
| | Walking |
| Activity participation | Daily/Saturday/Sunday routine |
| | Activities of daily living |
| | Leisure activity |
| | Social activity |
| | Productive activity (paid work) |
| | Education |
| | Rest and sleep |
| Nutrition | Rice or grains intake |
| | Bread or flour intake |
| | Potato or corn intake |
| | Meat or chicken breast intake |
| | Fish or tofu intake |
| | Beans or egg intake |
| | Sesame oil intake |
| | Nuts intake* |
| | Butter or margarine intake |
| | Green vegetables or kim-chi intake |
| | Fruit intake* |
| | Seaweed intake |
| | Dairy product intake* |
| | Cheese intake |
| | Anchovies intake |
| | Water intake |
| | Smoking habit |
| | Alcohol intake |
| | Protein intake for a week |
| | Carbohydrate intake for a week |
| | Fat intake for a week |
| | Vitamin intake for a week |
| | Mineral intake for a week |
| | Water intake for a week |

* Newly added items in Round 1.

## Discussion

The need to develop a multifaceted lifestyle profile for the elderly population has been identified in the literature; such a profile will support the successful aging and health of the older adults. South Korea is getting older, and the prevalence of disease and disability among the aging population is growing [21]. Previous research shows that lifestyle factors affect successful aging; thus, it is possible to extend the health span of the older adults [22]. Moreover, a large body of research and theoretical literature demonstrates that physical, cognitive, and social

**Table 4. Contents of the multifaceted lifestyle in elderly profile in the round 2 survey.**

| Category | Items | Sub-items | CVR* | Mean** | SD*** |
|---|---|---|---|---|---|
| Physical activity | Aerobic physical exercise | How many days did you do aerobic exercise in the last week? | 1 | 3.66 | 0.47 |
| | | On average, how many times did you do aerobic exercise during the day? | 0.90 | 3.58 | 0.64 |
| | | Do you do as much as you want? | 0.80 | 3.5 | 0.64 |
| | Anoxic physical exercise | How many days did you do anoxic physical exercise in the last week? | 0.90 | 3.5 | 0.64 |
| | | On average, how many times did you do anoxic physical exercise during the day? | 0.90 | 3.41 | 0.64 |
| | | Do you do as much as you want? | 0.71 | 3.16 | 0.68 |
| | High-intensity physical exercise | How many days did you do high-intensity physical exercise in the last week? | 0.80 | 3.58 | 0.64 |
| | | On average, how many times did you do high-intensity physical exercise during the day? | 0.71 | 3.5 | 0.76 |
| | | Do you do as much as you want? | 0.80 | 3.5 | 0.64 |
| | Moderate-intensity physical exercise | How many days did you do moderate-intensity physical exercise in the last week? | 0.80 | 3.5 | 0.64 |
| | | On average, how many times did you do moderate-intensity physical exercise during the day? | 0.80 | 3.41 | 0.64 |
| | | Do you do as much as you want? | 0.71 | 3.25 | 0.72 |
| | Low-intensity physical activity | How many days did you do low-intensity physical activity in the last week? | 0.61 | 3.33 | 0.74 |
| | | On average, how many times did you do low-intensity physical activity during the day? | 0.52 | 3.25 | 0.82 |
| | | Do you do as much as you want? | 0.52 | 3.08 | 0.75 |
| | Walking | How many days did you go walking in the last week? | 0.90 | 3.58 | 0.64 |
| | | On average, how many times did you walk during the day? | 1 | 3.75 | 0.43 |
| | | Do you do as much as you want? | 1 | 3.58 | 0.49 |
| Activity participation | Routine | Please write out your daily routine | 1 | 3.66 | 0.47 |
| | | Please write out your Saturday routine | 0.80 | 3.2 | 0.6 |
| | | Please write out your Saturday routine | 0.71 | 3.2 | 0.67 |
| | Activity of Daily Living (ADL) | How many days did you participate in ADL in the last week? | 0.90 | 3.58 | 0.64 |
| | | On average, how many times did you do ADL per day? | 0.90 | 3.5 | 0.64 |
| | | Do you participate in ADL as much as you want? | 1 | 3.66 | 0.47 |
| | Leisure activity | How many days did you participate in leisure activities in the last week? | 1 | 3.83 | 0.37 |
| | | On average, how many times did you do leisure activities per day? | 0.90 | 3.58 | 0.64 |
| | | Do you participate in leisure activities as much as you want? | 0.90 | 3.66 | 0.62 |
| | Social activity | How many days did you participate in social activities in the last week? | 1 | 3.66 | 0.47 |
| | | On average, how much time did you spend on social activities per day? | 0.90 | 3.5 | 0.64 |
| | | Do you participate in social activities as much as you want? | 0.90 | 3.58 | 0.64 |
| | Productive activity (paid work) | How many days did you participate in productive activities in the last week? | 0.90 | 3.58 | 0.64 |
| | | On average, how much time did you spend on productive activities per day? | 0.80 | 3.41 | 0.75 |
| | | Do you participate in productive activities as much as you want? | 0.80 | 3.5 | 0.64 |
| | Education | How many days did you participate in education in the last week? | 1 | 3.5 | 0.5 |
| | | On average, how much time did you spend on education per day? | 0.90 | 3.41 | 0.64 |
| | | Do you participate in education as much as you want? | 0.71 | 3.41 | 0.64 |
| | Rest and sleep | On average, how much time do you spend asleep per day? | 1 | 3.75 | 0.43 |
| | | Do you sleep as much as you want? | 0.90 | 3.66 | 0.62 |

(*Continued*)

**Table 4.** (Continued)

| Category | Items | Sub-items | CVR* | Mean** | SD*** |
|---|---|---|---|---|---|
| Nutrition | Protein | Have you eaten rice or grain each day? | 1.00 | 3.50 | 0.50 |
| | | Have you eaten bread or flour each day? | 1.00 | 3.42 | 0.49 |
| | | Have you eaten potato or corn each day? | 0.81 | 3.33 | 0.62 |
| | Carbohydrate | Have you eaten meat or chicken breast each day? | 0.90 | 3.42 | 0.64 |
| | | Have you eaten fish or tofu each day? | 0.90 | 3.42 | 0.64 |
| | | Have you eaten beans or egg each day? | 0.90 | 3.42 | 0.64 |
| | Fat | Have you eaten sesame oil each day? | 0.90 | 3.42 | 0.64 |
| | | Have you eaten nuts each day? | 1.00 | 3.50 | 0.50 |
| | | Have you eaten butter or margarine each day? | 0.71 | 3.25 | 0.60 |
| | Vitamins | Have you eaten fruit each day? | 0.90 | 3.50 | 0.50 |
| | | Have you eaten seaweed each day? | 0.90 | 3.42 | 0.64 |
| | Calcium | Have you eaten dairy products each day? | 0.71 | 3.25 | 0.83 |
| | | Have you eaten anchovies each day? | 1 | 3.33 | 0.47 |
| | | Have you eaten cheese each day? | 0.52 | 3.05 | 0.86 |
| | Minerals | How much water do you drink per day? | 0.81 | 3.42 | 0.76 |
| | Smoking | How much do you smoke per week? | 0.90 | 3.58 | 0.64 |
| | Alcohol | How often do you drink alcohol on average per week? | 1 | 3.67 | 0.47 |
| | | How much do you consume when you drink alcohol each time? | 1 | 3.67 | 0.47 |
| | Awareness of personal diet | How much protein do you think you consumed in the last week? | 0.81 | 3.17 | 0.90 |
| | | How much carbohydrate do you think you consumed in the last week? | 0.81 | 3.17 | 0.90 |
| | | How much fat do you think you consumed in the last week? | 0.81 | 3.08 | 0.86 |
| | | How many vitamins do you think you consumed in the last week? | 0.71 | 3.08 | 0.86 |
| | | How many minerals do you think you consumed in the last week? | 0.71 | 3.08 | 0.86 |
| | | How much water do you think you consumed in the last week? | 0.81 | 3.08 | 0.86 |

*CVR = Content Validity Ratio of all sub-items;

**Mean = average values of each sub-items;

***SD = standard deviation of each sub-items.

functioning are key elements of successful aging and that a multifaceted lifestyle influences them [23] [24]. Therefore, the interest in and demand to evaluate the lifestyles of the older adults are increasing worldwide. However, a review of the literature and previous studies showed that there are limited resources regarding multifaceted lifestyle profile assessment specific to community-based practice for the older adults [9]. The Delphi method was appropriate for surveying experts on this topic. Applying this method, we uncovered multidisciplinary recommendations for assessing the lifestyles of the elderly. We will discuss our suggestions based on the consensus of the expert panel in the study.

Previous research has shown that many age-related decreases in physical and cognitive functioning can be explained in terms of lifestyle factors such as physical activity [25]. Moreover, most national and international physical activity guidelines and position statements recommend that elderly individuals participate in regular physical activity as a means of preventing disease, promoting health, and delaying functional loss [26]. According to the expert panel, we also found that physical activities such as "aerobic and anoxic exercise" and "high-, moderate-, and low-intensity activity" are crucial factors in assessing the lifestyles of the older adults. In particular, in comparison with the existing lifestyle assessments, which evaluate low-intensity physical activities or walking, physical activities of various intensities

**Table 5. Contents of the multifaceted lifestyle in elderly profile in the round 3 survey.**

| Category | Items | Sub-items | CVR* | Mean** | SD*** |
|---|---|---|---|---|---|
| Physical activity | Aerobic physical exercise | How many days did you do aerobic exercise in the last week? | 1 | 3.79 | 0.41 |
| | | On average, how many times did you do aerobic exercise during the day? | 1 | 3.63 | 0.48 |
| | | Do you do as much as you want? | 0.8 | 3.47 | 0.68 |
| | Anoxic physical exercise | How many days did you do anoxic physical exercise in the last week? | 0.9 | 3.42 | 0.59 |
| | | On average, how many times did you do anoxic physical exercise during the day? | 0.8 | 3.21 | 0.61 |
| | | Do you do as much as you want? | 0.7 | 3.21 | 0.69 |
| | High-intensity physical exercise | How many days did you do high-intensity physical exercise in the last week? | 0.8 | 3.58 | 0.67 |
| | | On average, how many times did you do high-intensity physical exercise during the day? | 0.8 | 3.42 | 0.67 |
| | | Do you do as much as you want? | 0.8 | 3.32 | 0.65 |
| | Moderate-intensity physical exercise | How many days did you do moderate-intensity physical exercise in the last week? | 0.9 | 3.47 | 0.60 |
| | | On average, how many times did you do moderate-intensity physical exercise during the day? | 0.8 | 3.26 | 0.64 |
| | | Do you do as much as you want? | 0.7 | 3.16 | 0.81 |
| | Low-intensity physical activity | How many days did you do low-intensity physical activities in the last week? | 0.7 | 3.37 | 0.74 |
| | | On average, how many times did you do low-intensity physical activities during the day? | 0.7 | 3.26 | 0.71 |
| | | Do you do as much as you want? | 0.7 | 3.21 | 0.69 |
| | Walking | How many days did you go walking in the last week? | 1 | 3.79 | 0.41 |
| | | On average, how many times did you walk during the day? | 0.9 | 3.68 | 0.57 |
| | | Do you do as much as you want? | 0.9 | 3.53 | 0.60 |
| Activity participation | Routine | Please write out your daily routine | 3.58 | 0.59 | 0.9 |
| | | Please write out your Saturday routine | 0.8 | 3.26 | 0.64 |
| | | Please write out your Sunday routine | 0.6 | 3.16 | 0.74 |
| | ADL | How many days did you participate in ADL in the last week? | 0.9 | 3.58 | 0.59 |
| | | On average, how much time did you spend on ADL per day? | 0.9 | 3.53 | 0.60 |
| | | Do you participate in ADL as much as you want? | 0.9 | 3.63 | 0.58 |
| | Leisure activity | How many days did you participate in leisure activities in the last week? | 1 | 3.79 | 0.41 |
| | | On average, how much time did you spend on leisure activities per day? | 0.9 | 3.53 | 0.60 |
| | | Do you participate in leisure activities as much as you want? | 0.9 | 3.63 | 0.58 |
| | Social activity | How many days did you participate in social activities in the last week? | 1 | 3.79 | 0.41 |
| | | On average, how much time did you spend on social activities per day? | 0.9 | 3.58 | 0.59 |
| | | Do you participate in social activities as much as you want? | 0.9 | 3.74 | 0.55 |
| | Productive activity (paid work) | How many days did you participate in productive activities in the last week? | 0.9 | 3.74 | 0.55 |
| | | On average, how much time did you spend on productive activities per day? | 0.8 | 3.47 | 0.68 |
| | | Do you participate in productive activities as much as you want? | 0.9 | 3.68 | 0.57 |
| | Education | How many days did you participate in education in the last week? | 0.8 | 3.42 | 0.67 |
| | | On average, how much time do you spend on education per day? | 0.9 | 3.47 | 0.60 |
| | | Do you participate in education as much as you want? | 0.9 | 3.47 | 0.60 |
| | Rest and sleep | On average, how much time did you spend asleep per day? | 0.9 | 3.74 | 0.55 |
| | | Do you sleep as much as you want? | 0.9 | 3.58 | 0.59 |

(*Continued*)

**Table 5.** (Continued)

| Category | Items | Sub-items | CVR* | Mean** | SD*** |
|---|---|---|---|---|---|
| Nutrition | Protein | Have you eaten rice or grain each day? | 1 | 3.58 | 0.49 |
| | | Have you eaten bread or flour each day? | 1 | 3.37 | 0.48 |
| | | Have you eaten potato or corn each day? | 0.9 | 3.26 | 0.55 |
| | Carbohydrate | Have you eaten meat or chicken breast each day? | 1 | 3.58 | 0.49 |
| | | Have you eaten fish or tofu each day? | 1 | 3.63 | 0.48 |
| | | Have you eaten beans or egg each day? | 1 | 3.63 | 0.48 |
| | Fat | Have you eaten sesame oil each day? | 1 | 3.26 | 0.44 |
| | | Have you eaten nuts each day? | 1 | 3.32 | 0.46 |
| | | Have you eaten butter or margarine each day? | 0.8 | 3.16 | 0.59 |
| | Vitamins | Have you eaten fruit each day? | 0.9 | 3.42 | 0.59 |
| | | Have you eaten seaweed each day? | 0.9 | 3.47 | 0.60 |
| | Calcium | Have you eaten dairy products each day? | 0.9 | 3.42 | 0.59 |
| | | Have you eaten anchovies each day? | 0.9 | 3.37 | 0.58 |
| | | Have you eaten cheese each day? | 0.6 | 3.00 | 0.79 |
| | Minerals | How much water do you drink per day? | 0.9 | 3.53 | 0.60 |
| | Smoking | How much do you smoke per week? | 1 | 3.58 | 0.49 |
| | Alcohol | How often do you drink alcohol on average each week? | 1 | 3.68 | 0.46 |
| | | How much do you consume when you drink alcohol each time? | 0.9 | 3.63 | 0.58 |
| | Awareness of personal diet | How much protein do you think you consumed in the last week? | 0.8 | 3.32 | 0.65 |
| | | How much carbohydrate do you think you consumed in the last week? | 0.8 | 3.32 | 0.65 |
| | | How much fat do you think you consumed in the last week? | 0.8 | 3.32 | 0.65 |
| | | How many vitamins do you think you consumed in the last week? | 0.8 | 3.26 | 0.64 |
| | | How many minerals do you think you consumed in the last week? | 0.8 | 3.26 | 0.64 |
| | | How much water do you think you consumed in the last week? | 0.9 | 3.37 | 0.58 |

*CVR = Content Validity Ratio of all sub-items;

**Mean = average values of each sub-items;

***SD = standard deviation of each sub-items.

should be considered in order to provide recommendations on modifying the older adults' lifestyles in the future. The literature illustrated that consistent participation in moderate physical activity appears necessary to optimize physical and mental health [27] [28]. Thus, it seems that various types of physical activity should be evaluated when we develop a multifaceted lifestyle profile for the elderly.

The pattern of activity participation in daily life is also an important factor. Research has demonstrated that active participation in social activities [29] [30] [31], leisure activities [32], or productive and educational activities [33] [34] is able to enhance cognitive functioning [35]

**Table 6. Average opinions of the expert panel (n = 21).**

| | M* | SD** | CVR*** | Consensus |
|---|---|---|---|---|
| 2nd Delphi | 3.37 | 0.62 | 0.84 | 0.69 |
| 3rd Delphi | 3.46 | 0.59 | 0.87 | 0.76 |

*Mean = average values of all sub-items;

**SD = standard deviation of all sub-items;

***CVR = Content Validity Ratio of all sub-items.

[33], improve mental health [36], reduce functional disabilities [31], and delay mortality [37]. Although there is evidence that engagement in a variety of activities affects health outcomes, few studies have examined, including activity participation along with other lifestyle factors such as physical activity and nutrition . . . In this study, the expert panel agreed on the relevance and importance of measuring activity participation as one of the multifaceted lifestyle domains. Occupational balance has been shown to be related to health and well-being and it is a recurring crucial element in the occupational therapy literature as well as in health promotion [38] [39] [40]. Moreover, participation in various activities is associated with life satisfaction. Havighurst (1963) pointed out that people experience greater satisfaction with life by maintaining various activities and social roles as opposed to living a solitary lifestyle [41].

Older adults are particularly vulnerable to malnutrition [42]. As we get older, our bodies have different needs; thus, certain nutrients become especially vital for good health. Therefore, it is important for health professionals to investigate the elderly's daily diet; amount of food consumed; and balance of major nutrients such as protein, carbohydrates, fat, vitamins, minerals, and water. Several experts suggested that the questions should be modified to provide more specific and elderly-friendly examples in each sub-item and we modified several sub-items in the nutrition part. The literature illustrates that eating a healthy, balanced diet is key for older people to have more active lives [43] [44]. Diets evolve over time, being affected by complex factors including social and economic elements that shape individual dietary patterns. The proportion of malnutrition in the elderly in South Korea decreased from 28.7% in 1998 to 1 in 6 people in 2015, but it is still very poor compared to other age groups. Thus, promoting healthy food consumption by the elderly requires the involvement of multiple sectors and stakeholders. In order to do that, detailed nutritional evaluations of people's lifestyles are necessary.

This study proposed a multifaceted lifestyle profile for enhancing elderly health and quality of life. The main strengths of our study are its responses from a panel of multidisciplinary health professionals and its good response rates. However, some limitations also need to be recognized. This study analyzed only the content validity of the items in the multifaceted lifestyle profile by the expert panel. Therefore, validity and reliability studies should be conducted in the future. Finally, only health-care professionals from South Korea were invited to participate; thus, our panel of experts was not international. Hence, this research might only represent a limited viewpoint.

## Conclusions

The present study used a modified Delphi method to develop an elderly lifestyle profile. The process resulted in a total of 62 items divided into 3 categories. This assessment can be useful in clarifying the elderly's lifestyle and helping to identify individuals who require more in-depth lifestyle modification and related interventions. However, individual's lifestyle tends to be affected by various environmental factors including geographic factor and cultural factor. Therefore, further research should be conducted with people who live in multiple ethno-geographic regions. Also, further research is now required to confirm the validity and reliability of this measurement.

## Supporting information

**S1 Data.**
(DOCX)

## Author Contributions

**Conceptualization:** Ji-Hyuk Park.

**Data curation:** Kang-Hyun Park.

**Formal analysis:** Kang-Hyun Park.

**Funding acquisition:** Ji-Hyuk Park.

**Writing – original draft:** Kang-Hyun Park.

**Writing – review & editing:** Ji-Hyuk Park.

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
