## [Decision Letter · Decision Letter 0]

25 Mar 2020

PONE-D-20-05504

Development of an elderly lifestyle profile: A Delphi survey of multidisciplinary health-care experts

PLOS ONE

Dear Dr. Park,

Thank you for submitting your manuscript to PLOS ONE. After careful consideration, we feel that it has merit but does not fully meet PLOS ONE’s publication criteria as it currently stands. Therefore, we invite you to submit a revised version of the manuscript that addresses all the points raised during the review process.

We would appreciate receiving your revised manuscript by May 09 2020 11:59PM. To enhance the reproducibility of your results, we recommend that if applicable you deposit your laboratory protocols in protocols.io, where a protocol can be assigned its own identifier (DOI) such that it can be cited independently in the future. For instructions see: http://journals.plos.org/plosone/s/submission-guidelines#loc-laboratory-protocols

We look forward to receiving your revised manuscript.

Kind regards,

Gianluigi Forloni

Academic Editor

PLOS ONE

Journal Requirements:

2. Please provide additional details regarding participant consent. In the ethics statement in the Methods and online submission information, please ensure that you have specified (1) whether consent was informed and (2) what type you obtained (for instance, written or verbal, and if verbal, how it was documented and witnessed).

Reviewers' comments:

Reviewer's Responses to Questions

**Comments to the Author**

1. Is the manuscript technically sound, and do the data support the conclusions?

Reviewer #1: Yes

Reviewer #2: Partly

2. Has the statistical analysis been performed appropriately and rigorously? 

Reviewer #1: Yes

Reviewer #2: I Don't Know

3. Have the authors made all data underlying the findings in their manuscript fully available?

Reviewer #1: Yes

Reviewer #2: No

4. Is the manuscript presented in an intelligible fashion and written in standard English?

Reviewer #1: Yes

Reviewer #2: No

5. Review Comments to the Author

Reviewer #1: The authors tried to answer an interesting research question about the lifestyle influence on aging quality , that is which factors must be considered very important among the various possible agents. They considered three areas (Physical activity, Activity participation, Nutrition ) and used a Delphi method to achieve the outcome, with open and closed questions proposed to a panel of 21 experts in three successive steps. Finally they proposed 62 items with a relatively high consensus from the experts. In general the method seems to be correctly implemented and the collected data support the conclusion. The areas and items derived largely from previous researches and meta-analysis by the same authors, published on peer review journals, but unfortunately in Korean not so accessible to most readers ( but there is a summary in English). Likewise many factors are strongly influenced by the ethno-geographical area like nutrition. The authors correctly reported these elements as limitations at the end of the discussion. In the conclusion I suggest indicating the need for studies that include multiple ethno-geographic areas.

Participants : are the 21 experts the same across the three steps of the Delphi consultation? does it mean that no one gave up participating in the three steps or that only those who participated to the end were included? It should be specified.

There are no comments on those who refused to participate (do they have different characteristics than the participants?)

are the participants partly the same as in the published preliminary study? if so it should be specified

Now some detailed comments:

98 “through a file attached to an email.” what is its content? is it explained in the next sentence?

109 “All open-ended

110 questions were included to ensure that the survey accommodated the opinions from the experts” : this sentence is not clear to me, it must be explained or rewritten.

126 “The

127 level of consensus was set to 80% of respondents indicating agreement [18]”. This statement is not supported by reference 18. The same for :

135 If the coefficient of variation is less than 0.5, no further Delphi investigation is

136 required, and if it is 0.5–0.8, it is relatively stable [18].

213 The literature illustrates that eating a healthy, balanced diet is key for older people to have more

214 active lives [43]. Are you sure you can't find a better reference for such a general and important statement? You mentioned one cross-sectional research on a limited number of Brazilians.

Table 4 & 5 : the title “mean” of the second column must be explained, as well as “ M” in table 6

References :

19 : this is the seminal paper from Lawshe, are you aware of the Wilson’s criticisms about it? F. Robert Wilson, Wei Pan & Donald A. Schumsky (2012) Recalculation of the Critical Values for Lawshe’s Content Validity Ratio, Measurement and Evaluation in Counseling and Development, 45:3, 197-210, DOI: 10.1177/0748175612440286

38 Erlandsson LK, Eklund M. Levels of complexity in patterns of daily occupations: Relationships to women’s : it must be completed

42 : the URL didn’t work. are you sure you have correctly reported it ?

I tried another WHO reference

World Health Organization Healthy Diet. [(accessed on 2020)]; Available online: https://www.who.int/news-room/fact- sheets/detail/healthy-diet.

Reviewer #2: Thank you for the opportunity to review this manuscript outlining the development of a lifestyle profile for older adults. This work has the potential to contribute to the development of a measurement tool to identify to older adults in South Korea who would benefit from a lifestyle modification intervention. While the manuscript is mostly well-written and organised, and is based on a sound review of the literature, I have some concerns with the methods and results. There are also minor concerns relating to the clarity of some sentences, and the terminology used for older adults. These are outlined below and if the authors are able to adequately address these issues the quality of the manuscript will be improved.

Major issues

1. METHODS

• P5, line 97: “… asked to read about the concept of multifaceted lifestyle components” – it would be helpful to the reader to have this information included as a supplementary file.

• P5, line 99: “ … an online document consisting of 65 items” – is this different to the information they were provided about the concept of multifaceted lifestyle components?

• P5, line 101: “A total of 65 items …” – this is repetitive of line 99. I suggest consolidating the information.

• P5, line 103: Figure 1 – Figure 1 outlines “Stage 1, 2, 3” and “First, Second and Third Delphi Survey”. However the text in the Methods uses “Round 1, 2, 3” and “first-round questionnaire” (line 104). It would be easier for the reader to follow if the Figure and the text used the same descriptive wording.

• P5, lines 110-113: “After finishing their part, the panel participants were e-mailed ……. before starting the study” – this information is already mentioned in the previous paragraph. Suggest consolidating it into the first paragraph of Methods as it provides a clear understanding of what participants were required to complete prior to the first survey/round 1.

• P6, line 120: Suggest the use of a different word to ‘valence’ as the meaning of this word in the sentence is ambiguous; maybe a simpler word that clearly describes what participants were forced to choose would be easier to read?

• P6, lines 123-124: This section is headed Round 3 and mentions that four questions were modified with no questions excluded from Round 2. However the next sentence then goes back to talking about what happened in Round 2 (three items modified and one added) – which is confusing to me and I wonder whether it would be better mentioned in the Round 2 section? I apologise if I am misunderstanding, but I find it difficult to follow.

• P7, line 133: “21 panels” – I think this should read 21 panel experts as 21 panels were not used in this study.

RESULTS

• P8, line 151: consensus was reached for all 18 physical activity domain items, but Table 1 includes only 6 physical activity items? It is not until the I got to Table 4 that it was clear that you were talking about sub-items of the 6 PA items. I think this could be more clearly explained to avoid confusing the reader.

• P10, Table 4: This table outlines the contents of the second Delphi survey and lists questions relating to 59 sub-items. However in the Methods (p6, line 117) the authors state that the 2nd survey consisted of 62 questions. This discrepancy needs to be addressed and clearly explained.

• P13, line 168: “none of the 62 items analysed” – Table 5 includes 60 questions (as one item was added from the second survey). The discrepancy between 62 and 60 items needs to be addressed, or more clearly explained.

• P16, Table 6: M, SD and CVR need to be spelled out in full within the table or the abbreviations included as a legend to the table.

Minor issues:

1. Terminology for ‘older adults’. The authors use different terms throughout the manuscript, including ‘older persons’, ‘older adults’ ‘elders’, ‘the elderly’. The generally accepted terms exclude ‘elders’ and include ‘older adults or individuals or people’. This may, however be different in different countries, including for South Korea. I suggest using the accepted appropriate terminology and using it consistently throughout the manuscript.

2. INTRODUCTION

The introduction is clearly and logically organised and well set within the current literature. A few minor suggestions may improve the readability:

• P3, lines 55-56: “Only a few measurements evaluate one’s health-related lifestyle. However, they tend to only include …” – the link between these two sentences is not clear and I suggest combining the sentences to improve the flow, e.g. only a few measurements evaluate one’s health-related lifestyle; these tend to only include ….:.

• P3, line 58: I think what the authors are trying to say is that there has been little specific measurement of individuals’ multifacted lifestyles? The way this sentence is currently written is confusing to me and relies on the reader knowing what the multifaceted ‘areas’ are. A description of multifaceted lifestyle is included in the following paragraph, but some clarity earlier in the introduction would be beneficial.

3. DISCUSSION AND CONCLUSION

The discussion is well organised and easy to follow. Some minor suggestions include:

• P17, line 190: “crucial factors in assessing the lifestyles …” – it should be clarified that this is according to the expert panel.

• P17, line 196: Suggested deleting the word ‘numerous’ and simply state that: research has demonstrated …

• P17, lines 200-201: “few studies have examined these factors in the elderly with other lifestyle factors” – are you trying to say that few studies have examined these factors combined with other lifestyle factors? If possible this needs to be more clearly stated.

• P.17, lines 202-203: “we were able to assess the life-balance” – you did not assess this, you assessed the opinions of experts and it would be clearer for the reader if you stated that clearly here.

4. ABSTRACT

• P.2, line 29: “… questions reconstructed regarding the evaluation items …” – this meaning of this sentence is not clear to me. I am not able to suggest changes, but it should be rewritten to convey its meaning more clearly.

6. PLOS authors have the option to publish the peer review history of their article (what does this mean?). If published, this will include your full peer review and any attached files.

Reviewer #1: Yes: Antonio Guaita

Reviewer #2: No

---

## [Author Response · Author response to Decision Letter 0]

28 Apr 2020

Dear Editor and reviewers,

We thank you and the reviewers for a thorough reading and constructive review of our manuscript and for the opportunity to revise and resubmit. We are pleased to submit the improved research article. We upload the 'response to reviewers' file. Please find our response to each point raised by academic editor and reviewers. 

Academic editor

Journal Requirements: 

When submitting your revision, we need you to address these additional requirements: 1. Please ensure that your manuscript meets PLOS ONE's style requirements, including those for file naming. The PLOS ONE style templates can be found at http://www.plosone.org/attachments/PLOSOne_formatting_sample_main_body.pdf and http://www.plosone.org/attachments/PLOSOne_formatting_sample_title_authors_affiliations.pdf

To address the editor’s concern, the manuscript was modified in order to meet the PLOS ONE’s style. Based on the PLOS ONE style templets, we modified title, author, affiliations formatting. We hope that this formatting is appropriate for the PLOS ONE requirement.

2. Please provide additional details regarding participant consent. In the ethics statement in the Methods and online submission information, please ensure that you have specified (1) whether consent was informed and (2) what type you obtained (for instance, written or verbal, and if verbal, how it was documented and witnessed).

In page 4, we added details about participant consent. Informed consent was obtained for the Delphi survey. We sent email to potential participants including invitation letter which include information of the survey and written consent form. All participants submitted written consent form through email. We describe this process clearly in 94-97 lines.

3. Please include additional information regarding the survey or questionnaire used in the study and ensure that you have provided sufficient details that others could replicate the analyses. For instance, if you developed a questionnaire as part of this study and it is not under a copyright more restrictive than CC-BY, please include a copy, in both the original language and English, as Supporting Information

In order to address this concern, we provided Delphi questionnaire in English and Korean version as supporting information. We hope that this material can help you to understand the research. 

I updated my information, so I register my ORCID ID in Editorial manager. Please let me know, if any other issue regarding this. 

Reviewer #1:

The authors tried to answer an interesting research question about the lifestyle influence on aging quality, that is which factors must be considered very important among the various possible agents. They considered three areas (Physical activity, Activity participation, Nutrition) and used a Delphi method to achieve the outcome, with open and closed questions proposed to a panel of 21 experts in three successive steps. Finally, they proposed 62 items with a relatively high consensus from the experts. In general, the method seems to be correctly implemented and the collected data support the conclusion. The areas and items derived largely from previous researches and meta-analysis by the same authors, published on peer review journals, but unfortunately in Korean not so accessible to most readers (but there is a summary in English). Likewise many factors are strongly influenced by the ethno-geographical area like nutrition. The authors correctly reported these elements as limitations at the end of the discussion. In the conclusion I suggest indicating the need for studies that include multiple ethno-geographic areas.

Like your advice, we also deeply agree with that lifestyle tends to be affected by various environmental issues. Thus, we also mentioned it in the conclusion (you might check additional description in 236, 237, and 238) and we also are planning to conduct further study which include people who live in multiple ethno-geographic areas. 

Participants : are the 21 experts the same across the three steps of the Delphi consultation? does it mean that no one gave up participating in the three steps or that only those who participated to the end were included? It should be specified.

There are no comments on those who refused to participate (do they have different characteristics than the participants?)

are the participants partly the same as in the published preliminary study? if so it should be specified

1) In the study, all of the 21 experts participated in the 1,2, and 3 round survey. Yes, there was no one to drop out in the three steps. In order to clear this point, we describe it in the demographic of the panel experts (line numbers are 148, and 149). When we conducted this research, we reminded people about this survey due date before one week, and encourage them to complete the survey until round 3. We thought that that is why we could obtain the all participants’ reply.

2) Initially, when we recruited professionals who participate in this study, we sent out a total of 30 invitations to potential participants. Among them, only 21 people sent the written consent form and they said that they want to participate in the study. Regarding people who refused to participate in our study, we did not send any other additional email. Thus, we could not obtain information regarding people who refused our survey. However, like you said, we also agree that it is important to know the reason and characteristics of people who refused the survey. Hence, if we conduct further study, we will investigate the different characteristics between participants and non-participants.

3) The participants are the same persons in the published preliminary study. 

Now some detailed comments:

98 “through a file attached to an email.” what is its content? is it explained in the next sentence?

The file which we attached to an email is questionnaire. In this file, lifestyle factors such as physical activity, nutrition and activity participation were described as key domains of lifestyle briefly. This is because we want to give background information to participants regarding lifestyle elements which were developed by previous study. And then in the next page, a total of 65 items were included in the same questionnaire. We think that “a file” might make you and other readers confuse. Thus, we decide to change the sentence from the “through a file attacked to an email” to “through a survey questionnaire attached to an email” in order to be clear (in line number 104). Also, we describe more additional information regarding the survey questionnaire in line 104, 105,106. 

109 “All open-ended

110 questions were included to ensure that the survey accommodated the opinions from the experts” : this sentence is not clear to me, it must be explained or rewritten.

In order to be clear, we change the sentence like following. 

“The survey was developed with open and closed questions to accommodate expert opinions.” (line number 

126 “The

127 level of consensus was set to 80% of respondents indicating agreement [18]”. This statement is not supported by reference 18. The same for :

135 If the coefficient of variation is less than 0.5, no further Delphi investigation is

136 required, and if it is 0.5–0.8, it is relatively stable [18].

We checked the reference and the sentence that you pointed out, you are right, there was a mistake to cite reference. The correct reference number was [20] which was describe about Delphi method. Thus we revised the reference number correctly from [18] to [20] (line numbers are 136 and 145)

213 The literature illustrates that eating a healthy, balanced diet is key for older people to have more

214 active lives [43]. Are you sure you can't find a better reference for such a general and important statement? You mentioned one cross-sectional research on a limited number of Brazilians.

As you recommended, we search again so as to find some better references. Thus, we find a reference which can support the sentence and add it (line number 225). 

[44] de Groot, Lisette CPMG, Verheijden Marieke W, de Henauw, Stefaan, Schroll, Marianne, & van Staveren, Wija A. Lifestyle, nutritional status, health, and mortality in elderly people across Europe: a review of the longitudinal results of the SENECA study. The Journals of Gerontology, Series A . 2004; 59(12), 1277–12784. https://doi.org/10.1093/gerona/59.12.1277

Table 4 & 5 : the title “mean” of the second column must be explained, as well as “ M” in table 6

In the table 4 and 5, each sub-item’s mean was provided separately. However, in Table 6, mean values of all sub-items was provided. That is the mean in table 6 was the mean of 62 items. Participants scored whether each sub-item is relevant or not. At this time, the score consists of a 4-point Likert scale, and the average value of all sub-items is given in the table as “Mean”. We indicated the mean of the “Mean” at the bottom of the table.

References :

19 : this is the seminal paper from Lawshe, are you aware of the Wilson’s criticisms about it? F. Robert Wilson, Wei Pan & Donald A. Schumsky (2012) Recalculation of the Critical Values for Lawshe’s Content Validity Ratio, Measurement and Evaluation in Counseling and Development, 45:3, 197-210, DOI: 10.1177/0748175612440286

We appreciate for your useful information. Actually, we didn’t know that Wilson’s criticisms about Lawshe, thus, we read the paper. By reading this, we aware of this and if we conduct further study, we will consider it. 

38 Erlandsson LK, Eklund M. Levels of complexity in patterns of daily occupations: Relationships to women’s : it must be completed

We apologized that it was our mistake. So, we revise the reference correctly.

38. Erlandsson LK, Eklund M. Level of complexity in patterns of daily occupations: Relationships to women’s well-being. Occup. Sci. 2006. 27-36. http://doi.org/10/1080/14427591.2006.9686568

42: the URL didn’t work. Are you sure you have correctly reported it?

I tried another WHO reference

World Health Organization Healthy Diet. [(accessed on 2020)]; Available online: https://www.who.int/news-room/fact-sheets/detail/healthy-diet.

 When we tried to connect again, the URL worked. Like below page. Please tried again, and if you could not access the page, please let us know. 

Reviewer #2: 

Thank you for the opportunity to review this manuscript outlining the development of a lifestyle profile for older adults. This work has the potential to contribute to the development of a measurement tool to identify to older adults in South Korea who would benefit from a lifestyle modification intervention. While the manuscript is mostly well-written and organised, and is based on a sound review of the literature, I have some concerns with the methods and results. There are also minor concerns relating to the clarity of some sentences, and the terminology used for older adults. These are outlined below and if the authors are able to adequately address these issues the quality of the manuscript will be improved.

Major issues

1. METHODS

• P5, line 97: “… asked to read about the concept of multifaceted lifestyle components” – it would be helpful to the reader to have this information included as a supplementary file.

In order to understand the reader, we provided a supplementary file (the file name is supplementary 2). This information regarding the concept for multifaceted lifestyle component was developed by our previous research (Park K, Han D, Park H, Ha S, Park. Pilot research for development of multi-faceted lifestyle profile components affecting health and quality of life: Delphi survey. Korean Journal of Occupational Therapy. 2019; 27(3): 105–120).

In order to make clear, we modified the sentence in the manuscript like follow (P5. line 104)

“Participants were asked to read about the concept of multifaceted lifestyle components that were developed for creating a lifestyle profile of elders through a file attached survey questionnaire attached to an email (S1 Appendix). In the survey questionnaire, physical activity, nutrition and activity participation are described as key domains of lifestyle based on the literature review and previous study.”

• P5, line 99: “ … an online document consisting of 65 items” – is this different to the information they were provided about the concept of multifaceted lifestyle components?

Yes, we sent a survey questionnaire which contain two part to the participants. First section in the survey questionnaire demonstrated about the concept of multifaceted lifestyle components that were developed for creating a lifestyle profile based on the previous research (Park et al., 2019). The aim of the first section was to help participants to understand lifestyle components. And second part was consisted of 65 items. Thus, there were two kind of different information included in a survey questionnaire. However, we also agreed that readers might be confused like you, thus, we will provide a survey questionnaire as a supplementary file.

• P5, line 101: “A total of 65 items …” – this is repetitive of line 99. I suggest consolidating the information.

In order to make clear the manuscript, we consolidated the line 99 and line 101 like follow:

“Participants were then shown the next page which was consisted of 65 items: 19 items in the “physical activity” domain, 21 in the “activity participation” domain, and 25 in the “nutrition” domain” (P5. Line 106-108).

• P5, line 103: Figure 1 – Figure 1 outlines “Stage 1, 2, 3” and “First, Second and Third Delphi Survey”. However, the text in the Methods uses “Round 1, 2, 3” and “first-round questionnaire” (line 104). It would be easier for the reader to follow if the Figure and the text used the same descriptive wording.

We deeply agreed with your opinion, thus we modified the figure 1 outline form “First, Second, and Third Delphi as well as we modified the manuscript (P5. Line 111).

• P5, lines 110-113: “After finishing their part, the panel participants were e-mailed ……. before starting the study” – this information is already mentioned in the previous paragraph. Suggest consolidating it into the first paragraph of Methods as it provides a clear understanding of what participants were required to complete prior to the first survey/round 1.

As your comment, there were some sentences which should be reorganized. So as to make the manuscript clearly, we consolidated the line 110-113 sentences to the “Delphi method procedures” paragraph (P5, line 110-115). 

• P6, line 120: Suggest the use of a different word to ‘valence’ as the meaning of this word in the sentence is ambiguous; maybe a simpler word that clearly describes what participants were forced to choose would be easier to read?

We apologize for confused you. We looked into the sentence and we found that ‘valence’ was our mistake. Thus, we change the sentence like following: 

“A neutral middle point was not included to force participants to provide their obvious opinions.” (P6, line 127)

• P6, lines 123-124: This section is headed Round 3 and mentions that four questions were modified with no questions excluded from Round 2. However, the next sentence then goes back to talking about what happened in Round 2 (three items modified and one added) – which is confusing to me and I wonder whether it would be better mentioned in the Round 2 section? I apologise if I am misunderstanding, but I find it difficult to follow.

We agreed that the sentences from line 123 to 124 could make the reader confused. Therefore, we re-writhed the paragraph. We moved the sentence which are taking about round 2 from the ‘Round 3’ section to the ‘Round 2’ section Thus, the final version is like follow:

“Round 2

The round 2 survey was developed based on the participants’ responses in the first round. The round 2 survey consisted of 59 closed-ended questions grouped into 3 categories (physical activities, activity participation, and nutrition). The participants received the survey via e-mail and were required to score the relevance of each proposed element using a four-point Likert-type scale (strongly relevant, relevant, not relevant, or strongly irrelevant). A neutral middle point was not included to force participants to provide their obvious opinions (Table 1). In round 2, three items were modified and one item was added. Finally, 62 items were composed and grouped into 3 categories

Table 1 is here

Round 3

Round 3 modified four questions and no questions from the second round were excluded. For 65 items, 80% agreement was reached. In the third round, we required the participants to rate the relevance of each item using a four-point Likert-type scale ranging from 1 (strongly irrelevant) to 4 (strongly relevant). The level of consensus was set to 80% of respondents indicating agreement [20]. “

• P7, line 133: “21 panels” – I think this should read 21 panel experts as 21 panels were not used in this study.

To further clarify the sentence, line 133 sentence was revised as follow.

“According to the criteria, the CVR values of all items were set to 0.42 for 21 panel experts in rounds 2 and 3.”

RESULTS

• P8, line 151: consensus was reached for all 18 physical activity domain items, but Table 1 includes only 6 physical activity items? It is not until the I got to Table 4 that it was clear that you were talking about sub-items of the 6 PA items. I think this could be more clearly explained to avoid confusing the reader.

Like your comment, we agreed that the sentence (line 151) could make the reader confused. As a result of discussing with co-author, we decided to the deleted the item numbers and modified the sentence like below. 

“For the physical activity and activity participation categories, consensus was reached on all items” 

• P10, Table 4: This table outlines the contents of the second Delphi survey and lists questions relating to 59 sub-items. However in the Methods (p6, line 117) the authors state that the 2nd survey consisted of 62 questions. This discrepancy needs to be addressed and clearly explained.

As you pointed out, there was some mistake as we described the results of round 1,2, and 3. Like you said, three sub-items were missing from the Table 4. (Please write out your Saturday routine, please write out your Sunday routine, and have you eaten cheese each day). We corrected Table 4 again(P.10). Thank you for your comment and we also sorry to confuse you. 

• P13, line 168: “none of the 62 items analysed” – Table 5 includes 60 questions (as one item was added from the second survey). The discrepancy between 62 and 60 items needs to be addressed, or more clearly explained.

There was a similar mistake above, Thus, we looked at the Table 5, and corrected it. In order to prevent more confusing, we checked every number in the paper. 

• P16, Table 6: M, SD and CVR need to be spelled out in full within the table or the abbreviations included as a legend to the table.

In order for the reader to understand the information in the table 6, we added the abbreviations of M, SD and CVR in the table 6. 

Minor issues:

1. Terminology for ‘older adults’. The authors use different terms throughout the manuscript, including ‘older persons’, ‘older adults’ ‘elders’, ‘the elderly’. The generally accepted terms exclude ‘elders’ and include ‘older adults or individuals or people’. This may, however be different in different countries, including for South Korea. I suggest using the accepted appropriate terminology and using it consistently throughout the manuscript.

We also agreed with that using different terms for older adults would be able to make confuse the reader. Hence, we selected ‘older adults’ as an appropriate terminology and we used it consistently throughout the manuscript instead of ‘older persons’, ’elders’, ’the elderly’.

2. INTRODUCTION

The introduction is clearly and logically organised and well set within the current literature. A few minor suggestions may improve the readability:

• P3, lines 55-56: “Only a few measurements evaluate one’s health-related lifestyle. However, they tend to only include …” – the link between these two sentences is not clear and I suggest combining the sentences to improve the flow, e.g. only a few measurements evaluate one’s health-related lifestyle; these tend to only include ….:.

We are appreciated for your suggestion. As you suggested, we combined the sentences as follow. 

“Only a few measurements evaluate one’s health-related lifestyle; these tend to only include dietary factors, physical exercise, smoking, and/or drinking habits.”

• P3, line 58: I think what the authors are trying to say is that there has been little specific measurement of individuals’ multifacted lifestyles? The way this sentence is currently written is confusing to me and relies on the reader knowing what the multifaceted ‘areas’ are. A description of multifaceted lifestyle is included in the following paragraph, but some clarity earlier in the introduction would be beneficial.

We also agreed that ‘areas’ could make the readers confuse thus, we changed the sentence as follow in order to be clear. 

“While lifestyles are important to improving the health and well-being of older adults, there has been little specific measurement of individuals’ multifaceted lifestyles” 

3. DISCUSSION AND CONCLUSION

The discussion is well organised and easy to follow. Some minor suggestions include:

• P17, line 190: “crucial factors in assessing the lifestyles …” – it should be clarified that this is according to the expert panel.

In order to be clarified the sentence, we modified it as below, 

“According to the expert panel, we also found that physical activities such as “aerobic and anoxic exercise” and “high-, moderate-, and low-intensity activity” are crucial factors in assessing the lifestyles of the older adults.”

• P17, line 196: Suggested deleting the word ‘numerous’ and simply state that: research has demonstrated …

As your suggestion, we had made the sentence simpler by deleting the word ‘numerous’

“Research has demonstrated that active participation in social activities.”

• P17, lines 200-201: “few studies have examined these factors in the elderly with other lifestyle factors” – are you trying to say that few studies have examined these factors combined with other lifestyle factors? If possible this needs to be more clearly stated.

Yes, as you said that, we tried to say that few studies have conducted activity participations combined with other lifestyle factors such as physical activity and nutrition. We realized that this sentence is not clear, so we changed it as below.

“Although there is evidence that engagement in a variety of activities affects health outcomes, few studies have examined, including activity participation along with other lifestyle factors such as physical activity and nutrition.”

• P.17, lines 202-203: “we were able to assess the life-balance” – you did not assess this, you assessed the opinions of experts and it would be clearer for the reader if you stated that clearly here.

As you pointed out, that sentence was not appropriate because it is not based on the results, thus we decided to delete it so as to avoid misunderstanding of the reader. 

4. ABSTRACT

• P.2, line 29: “… questions reconstructed regarding the evaluation items …” – this meaning of this sentence is not clear to me. I am not able to suggest changes, but it should be rewritten to convey its meaning more clearly.

We agreed that the sentence make the readers confused. Thus we deleted some unnecessary words and make it more clear sentence as below. 

‘The first survey gathered opinions using a mix of open- and closed-ended questions regarding items of the older adults' multifaceted lifestyle profile.’

---

## [Decision Letter · Decision Letter 1]

8 May 2020

Development of an elderly lifestyle profile: A Delphi survey of multidisciplinary health-care experts

PONE-D-20-05504R1

Dear Dr. Park,

We are pleased to inform you that your manuscript has been judged scientifically suitable for publication and will be formally accepted for publication once it complies with all outstanding technical requirements.

With kind regards,

Gianluigi Forloni

Academic Editor

PLOS ONE

Additional Editor Comments (optional):

Reviewers' comments:

Reviewer's Responses to Questions

**Comments to the Author**

1. If the authors have adequately addressed your comments raised in a previous round of review and you feel that this manuscript is now acceptable for publication, you may indicate that here to bypass the “Comments to the Author” section, enter your conflict of interest statement in the “Confidential to Editor” section, and submit your "Accept" recommendation.

Reviewer #1: All comments have been addressed

Reviewer #2: All comments have been addressed

2. Is the manuscript technically sound, and do the data support the conclusions?

Reviewer #1: Yes

Reviewer #2: Yes

3. Has the statistical analysis been performed appropriately and rigorously? 

Reviewer #1: Yes

Reviewer #2: Yes

4. Have the authors made all data underlying the findings in their manuscript fully available?

Reviewer #1: Yes

Reviewer #2: (No Response)

5. Is the manuscript presented in an intelligible fashion and written in standard English?

Reviewer #1: Yes

Reviewer #2: Yes

6. Review Comments to the Author

Reviewer #1: No other comments or suggestions. The article has greatly improved. The authors revised and modified the text and references correcting any errors and completing the incomplete ones. In the "conclusions" they reported the importance of the ethnic-geographical context for research like this, following the suggestion. The results are interesting and could be the basis for other research on the same topic.

Reviewer #2: Thank you for addressing the comments from the review process and clarifying queries and concerns raised. I feel that your manuscript is greatly improved. I have two comments arising from your revision which need to be corrected to improve the language used:

1. p6, line133: while the revised wording is better I feel it would be much clearer to simply state: a neutral middle point was not included to force participants to provide their opinion (or response). The wording obvious opinions is confusing.

2. p.19, line 221: the revised wording has improved the sentence, however I suggest removing the word 'including' to make it clearer - few studies have examined activity participation along with other lifestyle factors such as physical activity and nutrition.

7. PLOS authors have the option to publish the peer review history of their article (what does this mean?). If published, this will include your full peer review and any attached files.

Reviewer #1: Yes: Antonio Guaita

Reviewer #2: No

---

## [Editor Report · Acceptance letter]

15 May 2020

PONE-D-20-05504R1 

Development of an elderly lifestyle profile: A Delphi survey of multidisciplinary health-care experts 

Dear Dr. Park:

I am pleased to inform you that your manuscript has been deemed suitable for publication in PLOS ONE. Congratulations! Your manuscript is now with our production department. 

With kind regards,

on behalf of

Dr. Gianluigi Forloni 

Academic Editor

PLOS ONE